# Epidemiology of brucellosis in cattle and dairy farmers of rural Ludhiana, Punjab

**Hannah R. Holt**[1]*, **Jasbir Singh Bedi**[2], **Paviter Kaur**[3], **Punam Mangtani**[4], **Narinder Singh Sharma**[3], **Jatinder Paul Singh Gill**[2], **Yogeshwar Singh**[3], **Rajesh Kumar**[5], **Manmeet Kaur**[5], **John McGiven**[6], **Javier Guitian**[1]

**1** Veterinary Epidemiology, Economics and Public Health Group, Department of Pathobiology and Population Sciences, The Royal Veterinary College, Hertfordshire, United Kingdom, **2** School of Public Health and Zoonosis, Guru Angad Dev Veterinary and Animal Sciences University, Ludhiana, Punjab, India, **3** Department of Veterinary Microbiology, Guru Angad Dev Veterinary and Animal Sciences University, Ludhiana, Punjab, India, **4** Faculty of Epidemiology and Population Health, Department of Infectious Disease Epidemiology, London School of Hygiene and Tropical Medicine, London, United Kingdom, **5** School of Public Health, Postgraduate Institute of Medical Education and Research, Chandigarh, India, **6** OIE Brucellosis Reference Laboratory, FAO Collaborating Centre for Brucellosis, Department of Bacteriology, Animal & Plant Health Agency, Surrey, United Kingdom

* hholt@rvc.ac.uk

**Data Availability Statement:** Relevant data of this manuscript is available to other researchers upon clearance from ethical review boards. Researchers should contact RVC Publications Repository

## Abstract

Brucellosis is a zoonotic disease imposing significant impacts on livestock production and public health worldwide. India is the world's leading milk producer and Punjab is the state which produces the most cattle and buffalo milk per capita. The aim of this study was to investigate the epidemiology of bovine brucellosis to provide evidence for control of the disease in Punjab State, India. A cross-sectional study of dairy farms was conducted in humans and livestock in rural Ludhiana district using a multi-stage sampling strategy. The study suggests that brucellosis is endemic at high levels in cattle and buffalo in the study area with 15.1% of large ruminants testing seropositive and approximately a third of dairy farms having at least one animal test seropositive. In total, 9.7% of those in direct contact with livestock tested seropositive for *Brucella* spp. Persons that assisted with calving and/or abortion within the last year on a farm with seronegative livestock and people which did not assist with calving/abortion had 0.35 (95% CI: 0.17 to 7.1) and 0.21 (0.09 to 0.46) times the odds of testing seropositive compared to persons assisting with calving/abortion in a seropositive farm, respectively. The study demonstrated that persons in direct contact with cattle and buffalo in the study area have high risk of exposure to *Brucella* spp. Control of the disease in livestock is likely to result in benefits to both animal and public health sectors.

## Author summary

Brucellosis is a bacterial disease that causes production losses in livestock due to abortions, increased calving intervals and reduced milk production. The disease can also be transmitted to humans via direct contact with livestock when they give birth or abort or via consumption of unpasteurised dairy products. This study aimed to estimate the frequency

(PublicationsRepos@rvc.ac.uk) to request access with an accompanying explanation as to what the data will be used for.

**Funding:** The project was enabled by joint funding from Biotechnology and Biological Sciences Research Council, UK (https://bbsrc.ukri.org/) and the Department of Biotechnology in India (http://dbtindia.gov.in/). Funding was awarded to JG and JM (BB/L004836/1), PM (BB/L004895/1) and NSS, JPSG, JSB, RK (BT/IN/Indo-UK/FADH/51/NSS/2013). The funders had no role in study design, data collection and analysis, decision to publish, or preparation of the manuscript.

**Competing interests:** The authors have declared that no competing interests exist.

of exposure to the bacteria (*Brucella* spp.) in cattle and buffalo in Ludhiana district, Punjab State, India. In addition, persons in contact with these livestock, either through their occupation or household, were also tested for *Brucella* spp. antibodies and the data used to identify factors increasing their risk of exposure. The study found high levels of exposure in cattle and buffalo in the study area with 15.1% of large ruminants testing positive for antibodies to *Brucella* spp. (seropositive) and approximately a third of dairy farms having at least one animal test seropositive. Around 10% of people tested seropositive for the bacteria and those that assisted with calving and/or abortion on a farm where at least one animal tested seropositive had a high risk of infection. Control of brucellosis could reduce production losses in livestock and protect humans in contact with livestock from becoming infected.

## Introduction

India is currently the world's leading milk producer; this study was conducted in Punjab which has the highest per capita milk production of all the Indian States (937g per person per day) [1]. Despite the development of cooperatives and increases in the number of organised dairy farms, a study in 2011 found the majority of milk in Punjab State was produced by rural smallholders and flowed through informal channels [2]. Endemic and emerging livestock diseases pose a threat to the milk industry due to reduced productivity of livestock (direct impact) and trade barriers (indirect). In addition, milkborne pathogens pose a threat to consumer health. Bovine brucellosis is a bacterial zoonosis of global importance due to its impact on livestock production, trade and human health [3,4]. In livestock, the disease reportedly causes economic losses to farmers through abortions and subsequently decreased milk yield [5,6].

Brucellosis is considered endemic in all states of India and recent increases in the incidence have been attributed to the intensification in the dairy industry resulting in increased cattle numbers and density as well as trade and livestock movement (Gwida, Al Dahouk et al. 2010, Kumar 2010). Herd-level and individual seroprevalence estimates in cattle and buffalo in Punjab have previously been estimated to be as high as 65.5% and 21.4%, respectively [7,8]. However, many of the previous studies have not used probabilistic sampling therefore the estimates may be biased. One study where sampling was unbiased was conducted over 10-years ago and reported a lower animal-level seroprevalence estimate (11.2%) [9].

Brucellosis is often cited as one of the world's most common bacterial zoonosis and is transmitted to humans via the oral, respiratory, or conjunctival routes either via direct contact or consumption of unpasteurised milk and dairy products [10,11]. Therefore, Incidence of the disease in humans relates to the frequency of brucellosis in the local livestock population and it is an occupational zoonosis for farmers and veterinarians [12,13]. The disease is underreported in many parts of the world, often misdiagnosed as malaria or typhoid due to non-specific clinical signs. According to the 'WHO estimates of the global burden of foodborne diseases: foodborne disease burden epidemiology reference group 2007–2015' there are 832,633 cases of brucellosis per year (95% Uncertainty Interval (UI): 337 929 to 19 560 440) [3]. However, in a systematic review of scientific studies estimating the frequency of human brucellosis (1990 to 2010) commissioned as part of this project no studies from the Asia region passed the validity assessment [14]. Given that Asia contains approximately 60% of the world's population, with India comprising around 17%, estimates of the frequency of human brucellosis here are critical to assess the global burden of the disease [15]. Knowledge of the relative importance of the different transmission routes can aid the design of surveillance systems and awareness

campaigns targeting human brucellosis. The WHO estimated that in 'South East Asia Region D', which contains India, the proportion of brucellosis cases via the foodborne route and direct contact with livestock is 0.45 (95% UI: 0.07 to 0.70) and 0.50 (95% UI: 0.22 to 0.82), respectively [3]. However, this estimate was based on the opinion of only seven experts and may have low precision, as reflected in the extremely wide uncertainty estimates. High quality data on the burden of human brucellosis in India are scarce. However, available evidence suggests humans are exposed via occupational contact in Punjab, with veterinarians and farm workers at high risk [12].

The objectives of this study were to estimate individual and farm-level seroprevalence of *Brucella* spp. in cattle and buffalo in dairy farms in Ludhiana district of Punjab and identify risk factors associated with *Brucella* spp. seropositivity. In addition, seroprevalence of *Brucella* spp. in persons in direct contact with these livestock was estimated and risk factor analysis performed.

## Methods

### Ethics statement

Ethical approval was obtained from the ethics committees at Guru Angad Dev Veterinary and Animal Sciences University, and the Post-Graduate Institute of Medical Education and Research (PGIMER), India, the London School of Hygiene and Tropical Medicine (LSHTM) and the Royal Veterinary College (RVC) in the UK. Informed, written consent for interviews and blood samples was obtained from all participants, or their legal guardian for minors.

### Study population and sampling

Between June 2015 and September 2017, a cross-sectional study of dairy farms was conducted using a multi-stage sampling strategy. For the purpose of the study, a 'dairy farm' was defined as any farm keeping at least four cows and/or female buffalo of reproductive age. Herds were classified as positive if at least one animal tested seropositive. Assuming an average of four lactating animals sampled per herd and an individual test specificity of 99% [16–19], herd specificity (HSp) was estimated to be 96%. Values of Herd Sensitivity (HSe) (likelihood of correctly classifying a positive herd) varied depending on herd size, number of lactating animals sampled and within-farm prevalence. Different combinations of these values were explored in order to achieve a desired HSe of 70% [20]. Using a design prevalence of 10% (the minimum within-farm prevalence to detect), sampling a total of nine animals per herd would result in HSe values above 70% given the values presented in Table 1. A sample size of 384 herds would allow herd-level prevalence to be estimated with a 6% precision and 95% confidence. Different precision values were trialled and 6% was selected as a trade-off between accuracy of the estimated and number of herds that could be sampled given available resources. This sample size was multiplied by an assumed design effect of 1.07, calculated from previous surveys, to account for correlation of herds within villages to give a final sampling target of 411 dairy farms.

In the first stage of sampling, four out of seven Community Development Blocks (CD block) (administrative division) of Ludhiana district were selected to restrict location for practical purposes. East Ludhiana has the largest human population [21], therefore this CD block was included and every other CD block was sampled working anti-clockwise in a systematic manner (Jagraon, Payal and Samrala). Within these CD blocks, 60 villages were randomly selected using sampling probability proportional to human size [21]. In order to meet the required sample size, eight dairy farms per village were selected using simple random sampling. In an average village, eight farms were estimated to comprise around 10% of the total

**Table 1. Values used to calculate sample size in order to estimate farm-level prevalence.**

| Variable | Value | Ref |
|---|---|---|
| Herd size | 5–20 | Pilot study |
| No. lactating animals sampled | 1–9 | Inputted value |
| Design prevalence (min) | 0.2 | Inputted value |
| Individual test sensitivity (milk ELISA) | 0.98 | [16–19] |
| Individual test specificity (milk ELISA) | 0.99 | |
| Calculated herd sensitivity (min) | 0.72 | http://epitools.ausvet.com.au/content.php?page= |
| Calculated herd specificity | 0.95 | HerdSens4 |
| Expected herd-prevalence | 0.25 | Expert opinion |
| Desired Precision (confidence) | 0.06 (0.95) | Inputted value |
| No. farms in study area | 10,000 | Calculated from Census of India. 2011 |
| Calculated sample size–unadjusted for clustering | 384 | Purposefully-designed spreadsheet |
| Design effect | 1.07 | Assumed from previous studies |
| Total herds–adjusted for clustering | 411 | |

dairy farms [21,22]. Lists obtained from the designated veterinary officer or pharmacist of each village were used as a sampling frame. If eight of fewer eligible farms were present in the study area then all farms were approached for inclusion in the study. If the dairy farm had one to nine lactating animals then all lactating animals were sampled where possible. If the dairy farm had more than nine animals, a systematic sample of nine animals was taken. Within selected farms, all people present involved with livestock husbandry (mostly farm owners, farm workers (employees) and family members) were offered serological screening for *Brucella* spp.

## Sample and data collection

From selected livestock, milk samples were collected from each quarter into a single 50ml poly-ethylene tube, after thoroughly cleaning and drying the teats. The samples were placed immediately into a cool box and refrigerated at 4˚C within five hours. Samples were aliquoted into Eppendorf tubes and these were then frozen at -20˚C for up to 6 months before de-frosting at room temperature for testing. Two 4ml venous blood samples were collected from each person selected using vacutainers one containing a clot activator and one containing EDTA. These were transported in a cool box to the School of Public Health and Zoonosis (GADVASU), centrifuged on the day of collection and stored at -20˚C.

Data regarding potential factors associated with *Brucella* spp. seropositivity in livestock were collected during face-to-face interviews using questionnaires pre-piloted in two villages (Questionnaires available on request). Causal diagrams to conceptualise potential pathways for farms, individual animals and persons in direct contact with livestock in the dairy farms being exposed to *Brucella* spp. were created (Fig 1). These diagrams were used to design the questionnaires and inform variable selection for statistical analysis. The first questionnaire was conducted with the farm owner or manager and collected data on potential farm-level risk factors including management practices and contacts with other livestock. The second, also conducted with the owner or manager, contained questions regarding individual characteristics of animals sampled. A third questionnaire was administered to persons in direct contact with livestock who were being tested for *Brucella* spp. and collected information of putative risk factors for human infection (livestock contact and dairy product) consumption. All interviews

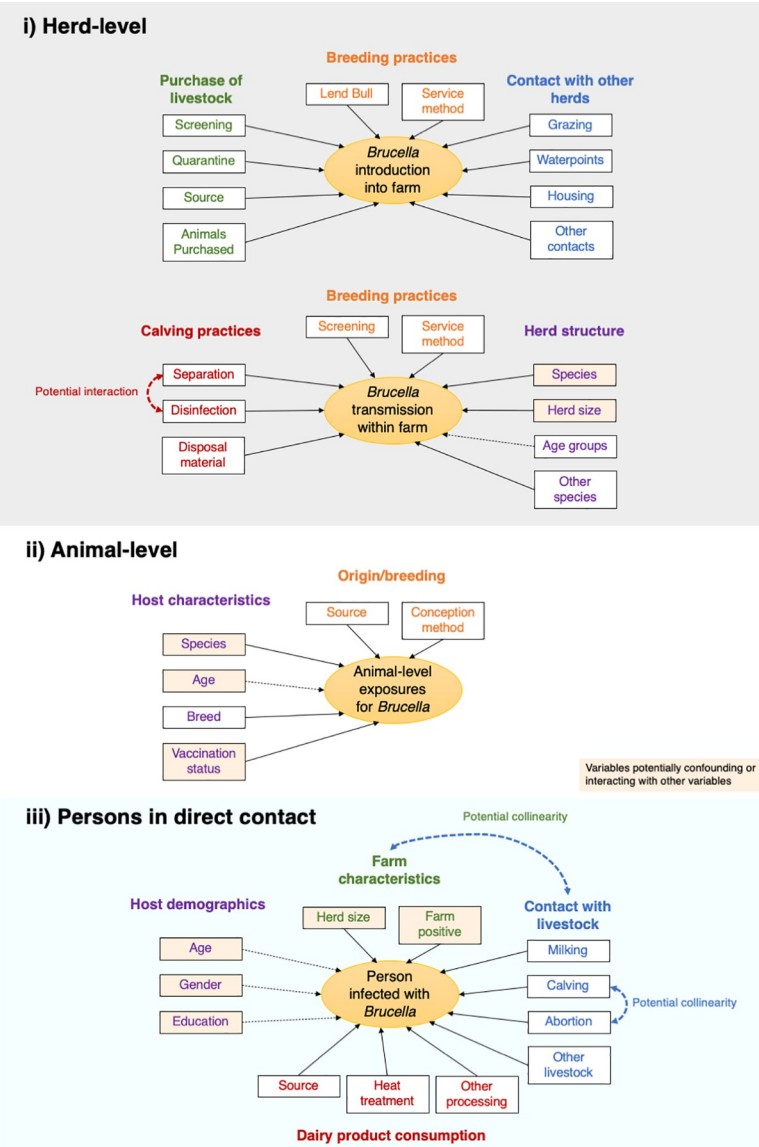

**Fig 1. Conceptual diagram depicting the variables hypothesised to be associated with seropositivity in i) dairy farms ii) livestock and iii) persons in direct contact with livestock in the dairy farms.** Data on these variables was gathered in the questionnaires and used in the statistical analysis.

were conducted in Punjabi by trained research fellows using paper forms with subjects anonymised and given a unique ID number.

## Serological testing

**Livestock samples.** Cattle and buffalo milk samples were screened for *Brucella* spp. using a commercial indirect milk ELISA developed by the OIE brucellosis reference laboratory at the Animal and Plant Health Agency (APHA), UK (BRUCELISA). The assay detects IgG antibodies to smooth *Brucella* strains in bovine milk samples. The BRUCELISA testing was carried out as per the manufacturer's instructions. A positive/negative cut-off was calculated as 50% of the mean of the optical density (OD) of the medium positive control wells. The medium

positive was used so that the positive/negative cut-off can be defined more robustly as it is closer on the dose-response curve to that cut-off than a strong positive control. Thus, potential inter-plate variation due to any fluctuation in the shape of the dose-response curve is minimised (Senior Scientist, APHA). Any test sample giving an OD equal to or above this was classified as positive. All milk samples which tested seropositive and a proportion of randomly selected negative samples were retested using the same procedure. All testing was performed in the Department of Microbiology, GADVASU.

**Human samples.** Human sera were tested using the Rose-Bengal test (RBT) for the detection of antibodies against *Brucella* spp. The antigen for this was supplied by the Punjab Vaccine Institute in India. Briefly, a sample of serum (0.05 ml) was mixed with 0.05 ml of antigen on a microscope slide to produce a zone approximately 2 cm in diameter. The mixture was agitated gently for four minutes at room temperature and observed for agglutination. If a visible reaction was observed, it was considered positive. A subset of samples (10 negatives, 7 positives) were also screened with antigen from the APHA for quality assurance. These samples showed perfect correlation. All serum samples were then tested for IgG antibodies for *Brucella* spp. using commercial ELISA kits supplied by DEMEDITEC Diagnostics GmbH, Kiel, Germany. As per the manufacturer's standard kit protocol, optical densities (ODs) were read at 450nm. For each plate, samples were interpreted as positive if ODs were >20% over the manufacturer's cut-off standard, negative as <20% under the cut-off, and inconclusive if in-between. All inconclusive ELISA samples were classified as negative for the purposes of this study. Both serological tests detect antibodies against the smooth lipopolysaccharide (sLPS) shared between the smooth strains of *Brucella* spp. (*B. abortus*, *B. melitensis* and *B. suis*). The Demeditec IgG iELISA (used in the serological testing of humans for prevalence of infection) was validated by the manufacturer though parallel testing against a CE-marked equivalent ELISA product. This study gave a value of 100% specificity (n = 88) and 100% sensitivity (n = 9).

All inconclusive ELISA samples were classified as negative. Additionally, all samples testing seropositive to the ELISAs or RBT were tested using the Standard Tube Agglutination Test (SAT) for diagnosis of active disease that may require treatment. Persons with a SAT titre of 1/160 International Units (IU) or higher received a phone-call from the principal investigator at GADVASU and were referred to a clinician at GADVASU who works with farmers and veterinarians to diagnose and treat brucellosis in the study area for further follow-up if they agreed.

## Statistical analysis

Seropositive samples from livestock were retested with the same BRUCELISA test and livestock were classified as positive if two seropositive test results were obtained. People were classified as seropositive if they tested positive to either RBT or IgG ELISA. Farms were classified as positive if at least one animal within the farm was classified seropositive for *Brucella* spp. after serial testing with the iELISA.

Univariate logistic regression analysis with random effects was used to provide crude estimates for the associations between candidate risk factors and *Brucella* spp. seropositivity in the three epidemiological units under study. Model outcomes were: i) individual animals testing seropositive (with farm and village included as a random effect), ii) dairy farms classified as seropositive (with village included as a random effect) and iii) persons in direct contact with livestock (with village included as a random effect). Farm was not included as a random effect for the model investigating risk factors in people in direct contact with livestock as only one person was tested in 75.7% of farms. Odds ratios (ORs) were calculated, comparing the odds of testing seropositive for *Brucella* spp. between each level of the variable and the baseline

category Variables where one or more categories were associated with *Brucella* spp. seropositivity with a *P*-value ≤ 0.2 were retained for further multivariate analysis.

Generalised linear mixed models (GLMM) were then used to identify risk factors for the three study units testing seropositive, controlling for potential confounders. Age was included as an *a priori* confounder in both the individual animal and human models. A manual forward step-wise procedure was used to build the models with variables with the lowest *P*-values entered into the model first. Upon addition of each variable, likelihood ratio tests were performed comparing the fit of the new model with the previous and to compare whether model fit was better when linear variables were kept as linear exposures or categorised. Variables were retained if there was significant evidence (*P*-value ≤ 0.0.5) against the null hypothesis that the simpler model was better and/or their addition changed (+/-) the OR for the association between another variable and *Brucella* seropositivity by ≥ 10% (possible confounding). Biologically plausible interaction terms were also investigated (Fig 1). Each model was checked for multicollinearity by calculating the Variance Inflation Factor (VIF) using the R package 'performance'; VIF values greater than 10 were considered non-tolerable. Odds ratios and 95% confidence intervals were reported. Intra-cluster correlation coefficients (ICCs) were also calculated from the final multivariate models. All analyses were performed in R version 3.5.2, GLMM was done using the package lme4.

## Results

### Livestock

The cross-sectional study included 413 farms in 58 villages (two of the villages did not have farms meeting the criteria for inclusion–at least four adult cattle and/or buffalo). In addition to producing milk for home consumption, the majority of farms sold milk to milk vendors in the informal sector (85.3%), whilst 44 (10.9%) sold to cooperatives, 6 (1.5%) sold to private companies and 9 (2.2%) kept all their milk for home consumption or sold to neighbours. Farm size ranged between 4 and 105 (Median: 10). Of the 1802 milk sample collected from cattle and buffalo, 312 initially gave a positive result and 272 produced two consecutive positive results using the commercial IgG ELISA (BRUCELISA). Therefore, individual livestock seroprevalence was estimated to be 15.1% (95% CI: 13.5 to 16.8). Overall, 96 (11.9%) buffalo and 174 (17.9%) milk samples from cows tested positive. A total of 136 farms had at least one animal that tested seropositive, therefore, farm-level seroprevalence was estimated to be 32.9% (95% CI: 28.6% to 37.6%).

**Dairy farms.** Only 10 (2.4%) farms kept goats and two of these had at least one large ruminant tested seropositive for *Brucella* spp. The results of the analysis for variables associated with farms testing seropositive are presented in Tables 2 and 3 (univariate).

In the final multivariate model, farms from Payal and East Ludhiana were more likely to be seropositive compared to Jagraon (Table 4). In addition, the odds of farms testing seropositive increased with the number of adult female cows on the farm. Unexpectedly, farms which reported "never" or "sometimes" disinfecting after calving had 0.43 (95% CI: 0.21 to 0.81) times the odds of testing seropositive for Brucella spp. The adjusted ICC of the farm model was 0.11, suggesting that 11% of the variance can be accounted for by the clustering of people within village.

**Individual livestock.** The results for the univariate analysis for individual livestock and farms testing seropositive for *Brucella* spp. is presented in S1 Table. Only 24 (1.3%) animals in four farms were vaccinated against *Brucella* spp. These animals were vaccinated using the reduced dose S19 vaccine via the conjunctival route. CD block, species, breed, origin and whether or not the animal was vaccinated against *Brucella* spp. were taken forward to the

**Table 2. Farm-level univariate analysis for associations between farm demographics and at least one animal in the herd testing seropositive using logistic regression with village included as a random effect.**

| Variable | Frequency (%) | No. Pos (%) | Odds ratio | P–value |
|---|---|---|---|---|
| **CD block** | | | | |
| Jagraon | 122 (29.5%) | 29 (23.8%) | 1 | - |
| East Ludhiana | 90 (21.8%) | 24 (26.7%) | 2.20 (1.26 to 3.89) | 0.006 |
| Payal | 113 (27.4%) | 46 (40.7%) | 2.33 (1.29 to 4.24) | 0.005 |
| Samrala | 88 (21.3%) | 37 (42.0%) | 1.17 (0.62 to 2.18) | 0.63 |
| **Total** | 413 | 136 | | |
| **Herd type** | | | | |
| Mixed | 243 (58.8%) | 69 (28.4%) | 1 | - |
| Buffalo only | 80 (19.4%) | 24 (30.0%) | 1.08 (0.61 to 1.86) | 0.783 |
| Cows only | 90 (21.8%) | 43 (47.8%) | 2.31 (1.40 to 3.81) | 0.001 |
| **Total** | 413 | 136 | | |
| **Total cows** | | | | |
| **None** | 107 (25.9%) | 32 (29.9%) | 1 | - |
| 1 or 3 | 129 (31.2%) | 35 (27.1%) | 0.87 (0.49 to 1.54) | 0.638 |
| 4 to 6 | 96 (23.2%) | 29 (30.2%) | 1.01 (0.55 to 1.85) | 0.963 |
| More than 6 | 81 (19.6%) | 40 (49.4%) | 2.29 (1.26 to 4.20) | 0.007 |
| **Total** | 413 | 136 | | |
| **Adult female buffalo** | | | | |
| None | 108 | 47 | 1 | |
| 1 or 5 | 204 | 58 | 0.52 (0.32 to 0.84) | 0.008 |
| More than 6 | 101 | 31 | 0.57 (0.32 to 1.01) | 0.056 |
| **Total** | 413 | 136 | | |
| **Total herd size** | | | | |
| 1 to 7 | 105 | 28 | 1 | - |
| 8 to 10 | 131 | 37 | 1.08 (0.61 to 1.94) | 0.787 |
| 10 to 15 | 106 | 36 | 1.41 (0.79 to 2.57) | 0.250 |
| More than 15 | 71 | 35 | 2.67 (1.42 to 5.09) | 0.002 |
| **Total** | 413 | 136 | | |

multivariate analysis. Only CD block and species was retained in the final model with age included as an *a priori* confounder (Table 5). Cows had 1.75 (95% CI: 1.29 to 2.43) times the odds of testing seropositive for the disease compared to buffaloes. Species and breed exhibited collinearity so only one was retained in the model. It was decided to keep species as there was some heterogeneity with how breed was recorded and some categories had few observations. The adjusted ICC for individual livestock was 0.164 for village and 0.005 for farm.

## Occupationally exposed survey

A total of 585 individuals which had direct contact with livestock in 360 (87%) of the studied farms gave informed consent to be screened for *Brucella* spp. and interviewed. The median number of persons sampled per farm was 1 (range 1 to 8). Based on a classification of IgG ELISA or RBT positive, seroprevalence (unadjusted) was estimated to be 9.7% (95% CI: 7.4% to 12.3%) with 57 people testing seropositive. Only 36 people (6.6%) had heard of brucellosis, with 6 (11.0%) of these testing seropositive. A total of 15 people thought they had suffered from brucellosis and 6 (40%) of these tested seropositive for the disease. Table 6 shows the comparison between results of the IgG ELISA and RBT. Only one sample was positive by RBT but negative by IgG ELISA, whereas 40 samples were positive by IgG ELISA and negative by

**Table 3. Herd-level univariate analysis for associations between herd management and at least one animal in the herd testing seropositive, using logistic regression with village included as a random effect.**

| Variable | Frequency (%) | No. Pos (%) | Odds ratio | P–value |
|---|---|---|---|---|
| **Purchased new livestock** | | | | |
| No | 369 (89.3%) | 124 (33.6%) | 1 | - |
| Yes | 44 (10.7%) | 12 (27.3%) | 0.74 (0.36 to 1.45) | 0.400 |
| Total | 413 | 136 | | |
| **Sold livestock** | | | | |
| No | 363 (90.3%) | 127 (34.0%) | 1 | - |
| Yes | 40 (9.7%) | 9 (22.5%) | 0.56 (0.25 to 1.17) | 0.144 |
| Total | 413 | 136 | | |
| **Insemination cows** | | | | |
| AI | 278 (88.3%) | 94 (33.8%) | 1 | - |
| Natural | 37 (11.7%) | 15 (40.5%) | 1.33 (0.65 to 2.67) | 0.420 |
| Total | 315 | 109 | | |
| **Insemination buffalo** | | | | |
| AI | 223 (74.1%) | 64 (28.7%) | 1 | - |
| Natural | 78 (25.9%) | 24 (30.8%) | 1.10 (0.62 to 1.92) | 0.729 |
| Total | 301 | 88 | | |
| **Natural service in either species** | | | | |
| No | 164 (67.5%) | 49 (29.9%) | 1 | - |
| Yes | 79 (32.5%) | 25 (31.6%) | 1.09 (0.60 to 1.93) | 0.779 |
| Total | 243 | 74 | | |
| **Disinfect after calving** | | | | |
| Always | 310 (77.3%) | 115 (37.1%) | 1 | - |
| Not always | 91 (22.7%) | 18 (19.8%) | 0.42 (0.23 to 0.72) | 0.002 |
| Total | 401 | 133 | | |
| **Separate at calving** | | | | |
| Always | 248 (62.5%) | 76 (30.6%) | 1 | - |
| Not always | 149 (35.7%) | 56 (37.6%) | 1.36 (0.89 to 2.98) | 0.156 |
| Total | 397 | 132 | | |

RBT. All samples which were inconclusive by IgG ELISA were classified as negative. Of the 57 people that were classified as seropositive, 13 (22.8%) produced SAT titres of 1/160 or higher (Table 7). All of these were RBT positive, however, only 2 mentioned symptoms in the last 12 months. See S2, S3 and S4 Tables for further description of the study population.

**Table 4. Herd-level multivariate GLMM for associations between farm-level risk factors farms testing seropositive for Brucella spp. using logistic regression with village included as a random effect.**

| Variable | Odds ratio (95% CI) | P-value |
|---|---|---|
| **CD block** | | |
| Jagraon | 1 | - |
| East Ludhiana | 2.62 (1.29 to 5.84) | 0.008 |
| Payal | 2.69 (1.19 to 5.91) | 0.015 |
| Samrala | 1.19 (0.53 to 2.74) | 0.671 |
| **Species** | | |
| Number of cows | 1.08 (1.03 to 1.15) | 0.006 |
| **Disinfect at calving** | | |
| Always | 1 | 0.01 |
| Not always | 0.42 (0.21 to 0.81) | |

**Table 5. Individual livestock multivariate GLMM for associations between individual level variables and cattle testing seropositive using logistic regression with village and farm included as a random effect.**

| Variable | Odds ratio (95% CI) | P-value |
|---|---|---|
| **CD block** | | |
| Jagraon | - | - |
| East Ludhiana | 2.48 (1.20 to 5.27) | 0.013 |
| Payal | 3.42 (1.59 to 7.49) | 0.001 |
| Samrala | 1.67 (0.78 to 3.66) | 0.174 |
| **Species** | | |
| Buffalo | - | - |
| Cow | 1.75 (1.29 to 2.43) | <0.001 |
| **Age** | 1.05 (0.98 to 1.13) | 0.165 |

CD block, gender, role on the farm, level of schooling and activities in the farm (milking; assisting with calving; assisting with abortion) were taken forward to the multivariate analysis. The variable 'how many abortions have you assisted with in the last 12 months' used in the statistical analysis (categorised as 0 or 'at least 1') as opposed to the categorical variable 'do you assist with abortions on the farm': yes (within the last 12 months); yes in the past; no. These variables were colinear, however, there was some misinterpretation in the latter question; some people reported assisting with abortion if it would be required as part of their role, even if they hadn't actually performed that task. No factors related to dairy product consumption were taken through to the multivariate analysis. Gender was not included as a potential confounder as there was collinearity between gender and the exposures due to gender roles within the farm. The ICC of the intercept-only model was 0.214, suggesting that 21.4% of the variation can be accounted for by the clustering of people within village. The results of the multivariate model are presented in Table 8. Assisting with calving and assisting with abortion were combined in to a new variable ("assisting with calving and/or abortion in the last 12 months") as GLMM did not converge when included separately in the final model. In the univariate analysis, people in direct contact with seropositive cattle and buffalo had 1.85 (0.98 to 3.50) times the odds of testing seropositive for *Brucella* spp. compared to those that did not (S5 Table). A biologically plausible interaction between farm status and association between assisting with calving/abortion and brucellosis status was investigated. However, only 5 people (5.7%) that did not assist with calving/abortion were seropositive (1 from a seropositive farm and 1 from a seronegative farm). As there were two few observations to include an interaction term between these variables, a new variable with three categories was created: i) assisted with calving and/or abortion in the last 12 months and from a farm classified as seropositive, ii) assisted with calving and/or abortion in the last 12 months and from a farm classified as seronegative and iii) have not assisted with calving or abortion in the past 12 months. Persons assisting with calving/abortion on a farm with seronegative livestock and people which did not assist with calving/abortion had 0.35 (95% CI: 0.17 to 7.1) and 0.21 (0.09 to 0.46) times the odds of testing seropositive compared to persons assisting with calving/abortion in a

**Table 6. Comparison of IgG ELISA and RBT results in persons in direct contact with large ruminants.**

| | | RBT | | Total |
|---|---|---|---|---|
| | | - | + | |
| **IgG ELSA** | - | 474 | 1 | **475** |
| | inc | 54 | 0 | **54** |
| | + | 40 | 16 | **56** |
| **Total** | | **568** | **17** | **585** |

**Table 7. SAT results of 57 persons in direct contact with large ruminants, classified as seropositive for Brucella spp.**

| Result | N | RBT + | IgG ELISA + | Symptoms? |
|---|---|---|---|---|
| ≥1/80 | 16 (28.1%) | 15 | 16 | 2 (1 fever, 1 joint pain) |
| ≥1/160 | 15 (26.3%) | 14 | 15 | 2 (1 fever, 1 joint pain) |
| ≥1/320 | 13 (22.8%) | 13 | 13 | 2 (1 fever, 1 joint pain) |

seropositive farm in the last 12 months. See S1 Text for full details on variables collected for human survey.

## Discussion

To our knowledge, this is the first epidemiological study of *Brucella* spp. in India where human and livestock populations in direct contact are studied concurrently, and one of only a few carried out in *Brucella* endemic settings [23–26]. The results suggest that brucellosis is endemic at high levels in cattle and buffalo of rural Ludhiana district in Punjab with animal-level seroprevalence estimated to be 15.1% (95% CI: 15.9 to 19.8). Two previous surveys conducted using random sampling in Punjab State more than 10 years ago estimated true-seroprevalence to be 11.2% and 17.8% [9,27]. Both these studies used indirect ELISA's with sLPS antigens, one on milk and one on serum. Further, approximately a third of dairy farms had at least one animal testing seropositive in this study.

This study also demonstrated that those in direct contact with large ruminants, either through their occupation or family herd, have a high risk of exposure to *Brucella* spp. with 9.7% of those screened testing seropositive. In a 'general population' survey of brucellosis where households in the same villages as the current study were randomly sampled (regardless of cattle ownership), 2.2% of persons screened were seropositive for *Brucella* spp. [22]. Risk factors for *Brucella* spp. seropositivity in this group were assisting with calving/abortion (61% of rural households kept cattle/buffalo) and consumption of goats' milk. Another previous study in Punjab screened veterinarians, veterinary pharmacists and animal handlers working for the Department of Animal Husbandry for *Brucella* spp. and found 21.9% had a positive RBT result, 24.0% had a positive STAT result, 19.7% had a positive IgM ELISA result, and 53.8% had a positive IgG ELISA result [12]. Although Proch et al. (2018) did not use probabilistic sampling, this further supports the notion that persons in direct contact with livestock in Punjab are at high risk of exposure to *Brucella* spp. Taken together, these results indicate that the disease poses a significant burden in rural Ludhiana, particularly in high risk occupations. The primary risk factor for testing seropositive for *Brucella* spp. in this survey was assisting with calving or abortion; cattle and buffalo infected with *Brucella* spp. excrete high

**Table 8. Final multivariate model for all variables associated with Brucella seropositivity in people in direct contact with livestock, including farm status, using logistic regression with village included as a random effect.**

| Variable | Odds ratio | |
|---|---|---|
| Assist with calving/abortion and farm status | | |
| Assist with calving/abortion on a farm with seropositive livestock | 1 | - |
| Assist with calving/abortion on a farm with seronegative livestock | 0.35 (0.17 to 0.71) | <0.001 |
| Do not assist with calving/abortion | 0.21 (0.09 to 0.46) | <0.001 |
| **Age** | | |
| Up to 30 | 1 | - |
| 31 to 40 | 1.10 (0.51 to 2.37) | 0.82 |
| >40 | 0.55 (0.26 to 1.15) | 0.11 |

concentrations of the organism in placental membranes and aborted foetuses. This effect depended on farms status; people that assisted with calving/abortion on a farm with seronegative livestock and people that did not assist with calving/abortion had 0.35 (95% CI: 0.17 to 7.1) and 0.21 (0.09 to 0.46) times the odds of testing seropositive compared to persons assisting with calving/abortion in a seropositive farm in the last 12 months. This strongly supports the existence of a relationship between assisting with calving and abortion of seropositive cattle and infection. The results of this survey, supported by the parallel survey in the general population suggest that there is limited exposure of *Brucella* spp. to people via milk and dairy products produced by large ruminants. This is likely due to the common practice of always boiling milk before consumption (85.1% in this survey) or as part of processing it into other dairy products. Given that approximately 20% of those that assisted with calving or abortion in a positive farm had evidence of exposure to *Brucella* spp., it is likely that transmission via this route is more important than the foodborne route in the study population. India has the largest bovine and second largest human population in the world. Although these results cannot be extrapolated, if the situation is similar in other Indian states producing high volumes of bovine milk, transmission of *B. abortus* via direct contact may be responsible for a significant burden of brucellosis infections in people.

Counterintuitively, farms that did not always disinfect after calving were less likely to test seropositive. It is unlikely that this is a true association and may be due to some unmeasured confounding as separation was more common in intensive dairy farms. Disinfection is only possible on certain surfaces, e.g. concrete floors, and it may be that animals which calved on surfaces that could not be disinfected (e.g. outdoors) were further away from the rest of the herd than animals which calved on surfaces where disinfection was possible. Husbandry was very similar in the majority of farms visited, which may explain the lack of other associated farm-level management factors.

## Study limitations

The study had several potential limitations. Only those persons having direct contact with livestock that were present at the time of the visit and gave informed consent were screened. However, eligible people were not formally enumerated nor refusals documented due to concerns farm workers may feel pressured by veterinary officers or their employers to take part. This could potentially bias the estimate if the characteristics of persons coming forward systematically differed from those who were not screened. However, this was considered unlikely given that people were not aware of brucellosis as a disease in humans and at least one person was recruited in the majority (87%) of farms using this passive recruitment method. Given the small herd sizes in the study area, it was often the case that one person was primarily responsible for the care of the animals, hence, only one person was recruited in the majority of farms. Only lactating animals were sampled, therefore male animals (which may play a role in disease transmission) were not tested, however, most farms were not keeping male livestock for breeding.

The S19 vaccine can interfere with serological tests [28]. Of the 28 animals vaccinated, 7 tested seropositive and it is unknown whether these were from infection (either prior to S19 conjunctival vaccination or because the vaccine does not confer full protection). Whether or not animals were vaccinated was not retained in the final model. If these animals were removed from the analysis then the animal seroprevalence results would reduce from 15.1% to 14.7% and farm-level seroprevalence would reduce from 32.9% to 32.2%. In addition, running logistic regression with vaccinated animals/farms removed led to the same variables being retained in the models. Therefore, if these were false positive results they do not appear to be biasing the results.

In the current study and the study by Proch et al. (2018) a large proportion of persons sero-positive by IgG ELISA did not test positive by RBT. The study in the general population also found poor agreement between the diagnostic test results in humans (Mangtani et al., 2020). This suggests there may be issues with the specificity of the ELISA used in these surveys, However, the results of the cross-sectional study and the parallel survey described in Mangtani et. al. (2020) show heightened exposure in persons in direct contact with livestock in the sampled dairy farms (9.7%) vs. general rural population (2.2%) and the risk factors identified correspond with the known biology of the disease, suggesting that misclassification due to low test performance may not have been an issue. Significant issues with the specificity of the iELISA would have likely resulted in more positive results in unexposed groups (false positives) than were observed. For example, of the 167 people who did not currently assist with calving and abortion, but had direct contact with livestock, only 5 (2.3%) were seropositive. Of these five people, three reported assisting with calving or drinking raw milk in the past. Compared to the superior situation for diagnosis in animals, there is a distinct lack of standardisation of serological reagents for diagnosis of human brucellosis.

## Recommendations

The best strategy for preventing introduction into negative dairy farms would be to screen new purchases, however, only four livestock owners reported doing this. The feasibility of this strategy depends on availability of screening tests for brucellosis at village level and farmers' capacity to afford this. According to this survey around a third of dairy farms already had at least positive animal in their herd, therefore, management strategies to reduce within-herd transmission of *Brucella* spp., particularly when animals are calving may be more effective. Although the majority of farms did report separating livestock at the time of calving (62.5%), they are often still in close proximity to the rest of the herd and are returned to the herd very soon after.

At the time of this study a programme termed '*Brucella* free village' was launched by the Department of Biotechnology with the mandate to "control and eradicate brucellosis in animals through stamping out of positive animals, vaccination of animals and maintaining sanitary conditions". The initiative is planned to be piloted across 50 villages in 10 states, including Punjab. The programme is very comprehensive planning to combine intensive screening, segregation of positives and vaccination. Although this will lower disease frequency in the selected villages there are concerns with the feasibility of upscaling this programme. The majority of the villages in this study would be classified as 'highly endemic' according to this programme's classification (prevalence >5%). The programme plans to isolate positive cows outside the village and slaughter positive buffalo. Given that almost 1 in 6 animals were seropositive in the cross-sectional survey, this strategy would require huge resources in this setting and may not be feasible unless prevalence was first sustainably lowered through methods such as vaccination.

A workshop was held at PGIMER campus on 21 November 2016 between State-level livestock, public health and food-safety actors around the theme of brucellosis control. The main objectives of the workshop were to ascertain what is being done to control brucellosis in Punjab State, identify opportunities to improve control, share project results with stakeholders and facilitate intersectoral communication. Nineteen participants attended the meeting, representing the main institutions involved in zoonosis control in Punjab, including Departments of Animal Husbandry, Dairy Development and Health and the Food and Drug Administration. In addition, public and animal health researchers, clinicians and representatives from the private sector (cooperatives and milk union) were present. Participants agreed that brucellosis

was an important animal and public health disease in the state. However, despite vaccination policies in place low coverage is achieved, this is supported by the findings of this study. Stakeholders attributed low coverage lack of vaccine availability and safety concerns of veterinarians and farmers. Therefore, these issues would need to be addressed before any vaccination campaigns are implemented.

As people are primarily infected via animal sources, control in livestock (primarily ruminants) is key to reducing the public health impact of the disease. However, whilst the seroprevalence remains high, it is recommended that the Dept. of Animal Husbandry works with health authorities to design public health campaigns which could reduce the risk of human exposure to brucellosis. Most people sampled said they were not aware of brucellosis as a disease in humans (93.4%). Further, through discussions during fieldwork and the stakeholder workshop, it seems that rural healthcare workers are also unaware of the disease. Therefore, it is likely that brucellosis goes unrecognised or misdiagnosed at the level of primary healthcare centres. Awareness was created in the sampled villages, however, there is a need to disseminate this information further. The issue of human diagnostics was also discussed during the workshop and clinicians raised highlighted the need to include brucellosis in the guidelines for investigating pyrexia of unknown. A technical expert group on human brucellosis was established as an output from the stakeholder workshop. This group is developing best practice guidelines for treatment and diagnosis and fostering awareness in the public health sector. Further as a result of this survey, samples are being sent to GADVASU from a tertiary hospital for screening.

As the primary risk factor was assisting with calving/abortion, public health campaigns should also include recommendations regarding the use of personal protective equipment (PPE) when assisting with calving. The final questionnaire for persons in direct contact with livestock did not contain questions on the use of PPE as results of the pilot surveys and experience of the field team suggest it is extremely rare for PPE to be worn whilst milking or assisting with parturition. This was observed during fieldwork and lack of PPE has been described in other brucellosis endemic settings [29–31]. Therefore, consideration as to whether this is a realistic strategy is needed. In addition to reducing the risk of exposure to the individual, appropriate hygiene and management of animals during calving can also reduce within-herd transmission of the disease in livestock. Clear messages backed by scientific evidence as to what farmers can do to protect themselves, their workers and their livestock from brucellosis are needed. These should be informed by assessments considering cost, feasibility and acceptability. Veterinary officers working in villages and the university extension services, which hold farmers' fayres and veterinary camps, are best placed to disseminate this knowledge. However, it is important that smallholders also receive these messages.

## Conclusions

Brucellosis is endemic at high levels in cattle and buffalo in the study area and the population is at risk of infection via direct contact. Appropriate control strategies to reduce production losses in livestock and prevent infections in those exposed to these livestock are needed.

## Supporting information

**S1 Table. Univariate analysis to identify factors associated with Brucella spp. seropositivity at animal-level.**
(DOCX)

**S2 Table. Association between demographic variables and Brucella spp. seropositivity in people in direct contact with large ruminants using univariable logistic regression models with village included as a random-effect.**
(DOCX)

**S3 Table. Association between livestock contact and Brucella seropositivity in in people in direct contact with large ruminants using univariable logistic regression models with village included as a random-effect.**
(DOCX)

**S4 Table. Association between dairy consumption and Brucella seropositivity in people in direct contact with livestock using univariable logistic regression models with village included as a random-effect.**
(DOCX)

**S5 Table. Association between data from livestock testing and farm questionnaire and Brucella seropositivity in people in direct contact with livestock using univariable logistic regression models with village included as a random-effect.**
(DOCX)

**S1 Text. Questionnaire used to interview persons in direct contact with livestock in the studied dairy farms.**
(DOCX)

## Acknowledgments

The authors would like to thank the research fellows that collected the samples for this project: Naresh Kumar and Sandeep Sodhi. We would also like to thank Lucy Duncombe and Anna Haughey from the Animal and Plant Health Agency for providing training in the application of the milk ELISA's utilised in this study. In addition, we would like to thank the veterinary offices and pharmacists working for the Department of Animal Husbandry in Punjab for their help facilitating the fieldwork and all the farmers that took part in the survey.

## Author Contributions

**Conceptualization:** Punam Mangtani, Narinder Singh Sharma, Jatinder Paul Singh Gill, Rajesh Kumar, Javier Guitian.

**Formal analysis:** Hannah R. Holt.

**Funding acquisition:** Punam Mangtani, Narinder Singh Sharma, Jatinder Paul Singh Gill, Rajesh Kumar, John McGiven, Javier Guitian.

**Investigation:** Hannah R. Holt, Jasbir Singh Bedi, Paviter Kaur, Yogeshwar Singh.

**Methodology:** Hannah R. Holt, Jasbir Singh Bedi, Paviter Kaur, Punam Mangtani, Narinder Singh Sharma, Jatinder Paul Singh Gill, Rajesh Kumar, Manmeet Kaur, John McGiven, Javier Guitian.

**Project administration:** Hannah R. Holt, Jasbir Singh Bedi, Paviter Kaur, Yogeshwar Singh.

**Resources:** John McGiven.

**Supervision:** John McGiven, Javier Guitian.

**Validation:** John McGiven.

**Writing – original draft:** Hannah R. Holt.

**Writing – review & editing:** Punam Mangtani, John McGiven, Javier Guitian.

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
