## [Decision Letter · Decision Letter 0]

20 Apr 2020

Dear Miss Holt,

Thank you very much for submitting your manuscript "Epidemiology of brucellosis in cattle and dairy farmers of rural Ludhiana, Punjab" for consideration at PLOS Neglected Tropical Diseases. As with all papers reviewed by the journal, your manuscript was reviewed by members of the editorial board and by several independent reviewers. In light of the reviews (below this email), we would like to invite the resubmission of a significantly-revised version that takes into account the reviewers' comments. 

We cannot make any decision about publication until we have seen the revised manuscript and your response to the reviewers' comments. Your revised manuscript is also likely to be sent to reviewers for further evaluation.

Sincerely,

Claudia Munoz-Zanzi

Associate Editor

Hélène Carabin

Deputy Editor

Reviewer's Responses to Questions

**Key Review Criteria Required for Acceptance?**

**Methods**

-Are the objectives of the study clearly articulated with a clear testable hypothesis stated?

-Is the study design appropriate to address the stated objectives?

-Is the population clearly described and appropriate for the hypothesis being tested?

-Is the sample size sufficient to ensure adequate power to address the hypothesis being tested?

-Were correct statistical analysis used to support conclusions?

-Are there concerns about ethical or regulatory requirements being met?

Reviewer #1: The creation of variables for analysis is unplanned. Many variables could have been merged to reduce the number of variables. The list/number of variables used to build models should be explicitly presented. 

Be consistent with the use of terms infected, positive and seropositive. Avoid using them interchangeably. Please double check everywhere in the manuscript.

East Ludhiana has the largest population: of animals or humans?

It is not clear why eight herds were selected in each village.

Serological testing: Do the serological tests used in the study detect antibodies against Brucella abortus or any Brucella spp? 

No results for SAT testing have been provided. Please clarify.

It is not clear if the same samples were retested or repeated samples were obtained from animals. The interpretation of results from these two scenarios would be different. Three sentences from different sections with different meanings are copied below. 

“All seropositive samples and a proportion of randomly selected negative samples were retested

using the same procedure.”

“Positive samples from livestock were retested and livestock were classified as positive if two

positive test results were obtained. People were classified as seropositive if they tested positive

to either the Rose-Bengal test or IgG ELISA.”

“Of the 1854 cattle and buffalo sampled, 280 tested positive on two consecutive milk ELISA’s,”

Change the sentence “Univariate analysis was performed” to “Univariate logistic regression analyses were performed”

Similarly, specify the model used for multivariable analyses: generalised linear mixed models (GLMM).

ICC: ICC can be calculated from GLMM, so you don't need to conduct ANOVA. Please refer to Vet Epi Research by Ian Dohoo, if not sure.

“Age was included as an a priori confounder” It is not clear if the human or animal age was included. Also, it is not clear in which model age was included as a confounder.

Were the assumptions of the model evaluated? If yes, how? Were the assumptions met?

Did you test interactions between significant variables in the model? It became apparent from the last model that the interactions were not tested. Please justify.

Did you test for potential collinearity between explanatory variables? If yes, how?

Reviewer #2: Please see attached document for comments.

Reviewer #3: -Are the objectives of the study clearly articulated with a clear testable hypothesis stated?

The objectives are given and are mostly clear. The third objective regarding human exposure risks could be more clearly stated (e.g. estimate prevalence and identify risk factors as for other populations rather than “exposure… was investigated” which is quite vague)

There is quite a lot of content in the results and more so in the discussion that is not very clearly aligned to these objectives so more focus and structure in the reporting of these sections can help improve this. 

-Is the study design appropriate to address the stated objectives?

The design reported sounds appropriate but some details are missing. 

For example

Can you clarify/explain your dairy farm definition – were farms keeping 4 cows AND/OR 4 female buffalo included? – you specify “OR” only currently but I think included mostly farms keeping both species. 

In the reporting of the multi stage selection can you reorder the content on Tehsil selection so that you explain ‘how’ selection was done at the beginning of section. 

You report ‘(Jagraon, Payal and Samrala)’ here without any explanation of what these are. 

The order of these section is not very clear. The sample size content could be moved to the beginning or end but appearing in the middle of the description of selection steps is quite confusing. Can you explain how you chose the value of 6% precision for this analysis as that’s not a standard default value? 

Can you give an indication of the denominators in each step? E.g. is 8 dairy farms per village a large of small % of the total? You say ‘up to eight dairy farms” were selected. When and why was this less than 8? 

If the farm had > 9 animals (NB – is this > 9 animals total or > 9 lactating females?) you report that a systematic sample was taken. How many animals is this? What criteria were sampling decisions made on? Was this really systematic sampling or convenience? Either is defendable – but it needs to be clear. The relevance of the statement re tethering is not clear. 

Can you give more rationale for the repeat testing approaches used and present the data on test repeatability for the different populations and tests in the results. Please report the results of the ‘validation’ of RBT antigens also – this can be in the SI but are important data to make accessible for evaluation. 

The level of detail given for different tests is variable – e.g. you report the cu-off rationale and rules for the BRUCELISA but not the DEMEDITEC test so this should be standardised (and explain how the kit defines inconclusives)

Please give more detail on the performance, results and guidance based on any SATs on human samples. 

The exact questions asked of respondents and the period of reference for the questions about risk behaviours should be included in the methods (the survey tool could be included as an SI file). 

-Is the population clearly described and appropriate for the hypothesis being tested?

The rationale for combining cattle and buffalo in several design and results reporting steps is not clear. What was the prevalence in each species? In the reporting of the model results the presence of different species and herd composition appear to be important but the detail on these metrics is not given or discussed very clearly. 

The composition of the human population reported on should be clearer. Can you give a clearer description of how this population was defined? Did you record data on the proportion of eligible individuals that ‘accepted’ the offer of screening? If not, can you estimate this? Assuming < 100% uptake what biases might this introduce and how does that affect your interpretation of the findings? 

In all of the tables and figures please include ensure that the population (animals, people or herds/farms) is clearly and simply stated each time – e.g. tables S5 and S6 are presumably about serostatus in humans but that isn’t stated. 

-Is the sample size sufficient to ensure adequate power to address the hypothesis being tested?

This calculation can be more clearly described. The content in the main paper is not very informative and I’d recommend stating the values currently reported in the SI in the text here so that all of the assumptions of the calculation are given up front. Please include a statement in the text of the sample size that the calculation indicated required (and then in the results, explicit statement that this was achieved)

-Were correct statistical analysis used to support conclusions?

The description and presentation of many of the stats and findings is often very descriptive and quite repetitive. You also report GLMM analyses that sound entirely appropriate for this dataset but then revert back to presentation and interpretation of raw data over the model outputs. 

This section would be clearer if reordered. Start with clear description of the populations and outcomes modelled. I think that a common approach was taken to model development for the 3 different populations. If so, compile all of the descriptions in one place and state this clearly. The description of the forward model building is not very clear. What order were variables evaluated and added in? Can you include references to support the decisions made based on coef and SE changes etc. 

You state that variables with a p<0.2 in univariable analyses were evaluated in multivariable models but its not clear what test the p value(s) mentioned are related to? If one level of a multifactor variable has a p for the coef estimate > 0.2 does that variable go through? Why were LRTs not used to evaluate the contribution to model fit of the variables as a whole?.

I think all models are logistic regressions with binomial errors but this is not clearly described (except in Table 3 legend). 

Many apparently continuous variables are categorised for model analyses and its not clear why? E.g. number of animals in different groups. Please explain or update as this seems to be an inefficient use of data. This also appears to change – in Table 3 the number of cows variable appears to be continuous? 

Can you explain justification of manual calculation of ICC metrics rather than reporting and using metrics derived from the random effects estimated in the models. Even if not used this way, all summaries of the GLMMs should include summary of the random effects estimated. 

Can you explain why the farm/herd RE was not included in the model of human serostatus data? Was this evaluated? This is presumably as you had many farms with 1 observation but should be stated clearly. 

-Are there concerns about ethical or regulatory requirements being met?

There is a mention in the methods that human serum agglutination tests were performed but no results are given on this. Were any presumptive cases identified? What case definition was used for this step and what procedures were followed for data sharing and clinical advice for individuals who were tested – and then received positive or negative results?

**Results**

-Does the analysis presented match the analysis plan?

-Are the results clearly and completely presented?

-Are the figures (Tables, Images) of sufficient quality for clarity?

Reviewer #1: ALL TABLES: 

- Please report overall p-values for each variable. These p-values should be used to make decisions about whether or not a variable is significant, instead of the p-values currently presented.

- Odds ratio for the reference category should be 1. Please replace ‘-‘ with 1 in all tables.

Table 1

- Not sure why cows and buffalo are two different variables. The variable herd type adequately captures this information. retain herd type and delete the other two variables.

- What is the reason for testing the variables cow and buffalo calves? Justify or exclude.

- Is there any reason for including number of cows and buffaloes separately, in addition to the variable total herd size? Justify or exclude (i.e. just retain total herd size).

Table 2

What is LR? Please explain the abbreviation in a footnote to the table.

Insemination cows, insemination buffalo, natural service: why are there three variables instead of one? The variable should be ‘type of insemination: natural or AI’ 

Table 3

- Please see my comment about overall p-values above.

- Why are the odds ratios for Sub-district very different from those of univariable results? For example, the odds of Samarala are nearly double than that of East Ludhiana in Table 1 and nearly half of East Ludhiana in Table 3. Please double check to make sure that you are using the same categories as in Table 1. Note that the odds ratios can change after the addition of other variables in the model, but such a drastic change needs to be investigated (e.g. the variables in the final model may be collinear). Also, the herd prevalence estimates in Table S2 do not seem to align with the estimates in Table 1. Please review these data and results.

- I assume that the ‘number of cows’ variable was used as a numeric variable. Did you test whether its association with the logit of the outcome linear?

- Add the reference category for the last variable in Table 3.

Table S3:

- Why are there two variables for ‘role on farm’ and religion? 

- It doesn’t seem appropriate to combine Muslims with Hindus just because their numbers are small. The proportions seropositive were actually similar in Sikh and Hindus, so you are going to make incorrect conclusions by clubbing Hindus with Muslims. 

Table S4: 

- It is not clear why odds ratios and p-values have been suppressed for some variables. 

- Why are there two variables for each of the following variables: ‘milking LR’, ‘assisting with calving’, ‘assisting with abortion? 

- 

Table 4

- Looks like age is used as a numeric variable here. Did you test the assumption of linearity for age?

- In contrast, in Table S4, age was included as a categorial variable. Please be consistent. Why was age categorised for univariable analyses and then used as numeric for multivariable analyses?

Table 5 is unnecessary. Just briefly present the results in the text.

Table 6

- Combining never with the past category is not advisable. 

- Which variables were tested in this model?

- Why not test if the interactions were significant, instead of arbitrarily combining variables? Delete Tables 8 and 9 and include the interaction term in Table 6 itself.

Section 2.1.2

- Please report ICC values for other models.

Table 6: Can you please clarify which variables were included in the model presented in Table 6? Were all the variables in tables from S3 to S5 tested in this model?

“Angandwadi workers”: Briefly describe who they are. Are they healthcare workers or medical practitioners?

Discussion, second last paragraph: Please define PPE.

Discussion, last paragraph: “Given the absence of PPE use in the current survey”. I might have missed but didn't notice this finding in the current study. Please clarify.

Reviewer #2: Please see attached document for comments.

Reviewer #3: -Does the analysis presented match the analysis plan?

The ordering of the results could be more clearly mapped to the study objectives – e.g. consistent ordering of animals, herds and humans. The livestock results sections opens with a description of milk consumption patterns which aren’t really mentioned before this point. Similar the dairy farms section is focused on the small n farms that kept goats. This is a confusing point to lead with as its not focused on any of the stated objectives, the data are very sparse and none of the results presented are informed by the model analyses performed. Several associations are reported without an explicit statement of (or reference to the table) the results from the analyses presented that support each statement. 

Its not clear why the data for cattle and buffalo are reported together vs by species. 

Much of the reporting of the univariable analyses can be dropped so that the results are focused on the multivariable model findings. All of the univariable results are given in the tables (I think) but these are not consistently presented. Some include indications of the variables evaluated in multivariable models but not all. The explanation of many variables is unclear – does Buffalo Yes/No mean indicator of species presence on the farm? 

There are several cases where a set of correlated variables are evaluated and its not clear how decisions were made on which variable to carry through to multivariable analysis? E.g. were model presence of buffalo and herd type fitted in the same models? 

-Are the results clearly and completely presented?

The levels of many factors are not really explained. Its also important to explain clearly how some of the factor levels are defined. Its not clear from the info given that the decision to combine Never and past and compare with “Yes” is a justifiable choice. What are the time periods for each level? Why 

Do you need to present both model 6 and 7? Why is age retained in the model? 

In some of the results test sections the results focused on are not very obviously focused on the objectives of the paper and involve quite a lot of anecdotal description of data, to suggest associations that were either not statistically evaluated or are not support by the models presented – e.g. symptom reporting in occupationally exposed period. Given the very low diagnostic value of clinical signs, use of serostatus vs clinical illness as the outcome, long persistence of Ab and retrospective nature of the survey there is no very strong reason to evaluate evidence of any association. These data are not mentioned elsewhere in the methods or results. 

You describe a slight increase of seroprevalence with age based on the figure but as the models indicate no association I don’t its justifiable to present this as an increase with age. 

Quite a lot of the descriptive text on e.g. milk consumption practices could be replaced by referring to the detail on these data (and the model evaluations performed) in a table (s6) rather than brief text summary. 

The interpretation of the data on risk factors for human exposure is complicated by the lack of clarity on the population sampled and the way the variables were evaluated. The rationale for the descriptive analysis of association between different explanatory variables is not clear. Why weren’t these effects evaluated using interaction terms? The nature of the comparator population in reporting of the increased risk in people with occupational exposure to seropositive animals is not clear. Its also unclear why a new model is used to evaluate this variable/interaction vs including the interaction in the full model for this population? 

-Are the figures (Tables, Images) of sufficient quality for clarity?

In the first figure can you update the legend to explain the overlap shading or update the plot so that the data for the sexes are presented adjacent but not overlapping. 

In the tables, more detail on the variables and levels presented are needed.

What is “LR”?

**Conclusions**

-Are the conclusions supported by the data presented?

-Are the limitations of analysis clearly described?

-Do the authors discuss how these data can be helpful to advance our understanding of the topic under study?

-Is public health relevance addressed?

Reviewer #1: The conclusions will change after analyses are repeated after finalising the variables and their categories.

Reviewer #2: Please see attached document for comments.

Reviewer #3: The correspondence in terms of structure and key points addressed between the discussion/conclusion and the rest of the paper is not very strong 

Some of the key findings are not really discussed – why do you think there is such a strong effect of region on serostatus? Why might this be? What is your interpretation of the ICC data presented?

There is quite a lot of content in the discussion about a sister paper that is not yet published. Its not really appropriate either present findings from another study here or to include reference to the other study until the other paper can be cited. 

-Are the conclusions supported by the data presented?

The speculation about infecting species of Brucella is not supported by any of the data presented in this paper and I would not include this for that reason

Some of the points about relative importance of different routes of transmission are interesting but don’t make as good use of the available data as is possible to support these points. 

The description of the counterintuitive finding is a little speculative. The argument that people might take more precautions during parturition if aware that there was brucellosis in their herd is somewhat undermined by the low levels of awareness of the disease that is presented elsewhere? 

The content on possible control, surveillance and transmission prevention strategies is interesting and important but is currently mentioned in a few different places and is not really linked to any of the data presented or clearly described in the context of options under consideration in this setting. What control options might be feasible in this context? How do the data presented here help guide thinking about these options? There is quite a lot of narrative content presented in this section describing findings from other studies (e.g. on PPE) that don’t currently add a lot to the interpretation of the findings from this study. 

-Are the limitations of analysis clearly described?

There is not an explicit section on study limitations

-Do the authors discuss how these data can be helpful to advance our understanding of the topic under study?

-Is public health relevance addressed?

The public health relevance of the findings is mentioned but the content is quite vague and non-specific. The general points about the challenges of vaccination are not really the findings of this study. Can you include more focused content on what the findings of this study tell us and how they could be used to guide or focus next steps research or pollcy?

**Editorial and Data Presentation Modifications?**

Reviewer #1: Abstract

Change “The aim of this study is to” to “The aim of this study was to”

Author summary

Reword this sentence: “This study to estimate the frequency of exposure to the bacteria

(Brucella spp.) in dairy animals in Ludhiana district in Punjab State of India and persons in

contact with these livestock either through their occupation or household.”

Introduction

Replace comma with a period: India is currently the world’s leading milk producer. This study was conducted…

Reword: produces the most cattle and buffalo’ milk…

Reword or split: Endemic and emerging livestock diseases pose a threat to the milk industry due to reduced productivity of livestock (direct impact), trade barriers and, if a milkborne zoonosis, posing a

threat to consumer health.

Corbel, 1997: Try to use primary sources if available.

Reword or split: However, many of the previous studies have not used

probabilistic sampling therefore the estimates may be biased, an example of `one study where

sampling was unbiased was conducted over 10-years ago and reported a lower animal-level

seroprevalence estimate. 

Reword or split: Incidence of the disease in

humans relates to the frequency of brucellosis in the local livestock population it is an

occupational zoonosis for farmers and veterinarians

Reword or combine: Only one sample was positive by

RBT but negative by IgG ELISA. Whereas 40 samples were positive IgG ELISA and negative by

RBT.

Split into two sentences: Only 24 (1.3%) animals in four farms were vaccinated against Brucella spp. this was mostly using the reduced

Reword or combine: The majority of the participants were male (69.1%) and over 20 (92.4%). With family members accounting for the majority of people in occupational contact with cattle on dairy farms (56.9%), with around owners accounting for 23.2% and employees accounting for 19.9%.

Reviewer #2: (No Response)

Reviewer #3: Please include page and line numbers for future submissions as that greatly improves the review process

There are several places where the terminology is inconsistent which creates ambiguity and confusion. Try to use a stripped back set of terms consistently - e.g. herds vs dairy farms vs farms, the descriptions of the various human populations need clarification throughout (people present ‘involved with husbandry’, household members, family members, farm workers, occupationally exposed group etc are used at various points and its not clear how these all relate to each other).

There are acronyms that need to be checked for full explanation at first use also.

**Summary and General Comments**

Reviewer #1: This study was conducted to estimate the prevalence of brucellosis in cattle, buffalo and in-contact humans and to identify the risk factors for seropositivity. The manuscript has some novel data about brucellosis epidemiology in India. The authors should be commended for using a robust design for the study, in particular, for using probabilistic sampling methods to select villages and animals. However, the casual approach to data analysis and writing the manuscript was disappointing and could have been improved. The manuscript has many grammatical errors, particularly, in the first two sections. I have listed some of the issues above, but the authors should read the manuscript thoroughly before resubmission. The manuscript should be of interest to the journal audience, but the authors would have to rebuild most of the models and thoroughly revise the manuscript before it could be accepted for publication.

Reviewer #2: Please see attached document for comments.

Reviewer #3: Overall this paper presents data from an interesting study in a context where data on brucellosis are limited. These data have definite value but are currently under-utilised and the presentation of the key findings from this work is not as clear as it should be. 

Some updates to the structure of the article can greatly increase the focus and clarity of the work presented. There are also updates needed to the presentation of the statistical analyses (see notes above).

PLOS authors have the option to publish the peer review history of their article (what does this mean?). If published, this will include your full peer review and any attached files.

Reviewer #1: No

Reviewer #2: No

Reviewer #3: No
---

## [Decision Letter · Decision Letter 1]

4 Jan 2021

Dear Miss Holt,

We are pleased to inform you that your manuscript 'Epidemiology of brucellosis in cattle and dairy farmers of rural Ludhiana, Punjab' has been provisionally accepted for publication in PLOS Neglected Tropical Diseases.

Best regards,

Claudia Munoz-Zanzi

Associate Editor

Hélène Carabin

Deputy Editor

Reviewer's Responses to Questions

**Key Review Criteria Required for Acceptance?**

**Methods**

-Are the objectives of the study clearly articulated with a clear testable hypothesis stated?

-Is the study design appropriate to address the stated objectives?

-Is the population clearly described and appropriate for the hypothesis being tested?

-Is the sample size sufficient to ensure adequate power to address the hypothesis being tested?

-Were correct statistical analysis used to support conclusions?

-Are there concerns about ethical or regulatory requirements being met?

Reviewer #1: Yes. The authors have appropriately revised the manuscript based on reviewers' suggestions. I think Table 1, 6 and 7 can be presented as supplementary materials but I will let the editor make a final decision.

Figure 1 can also be presented as supplementary material. Regardless, I don't think it is appropriate to call it a causal diagram. I would be better to refer to it as a 'conceptual diagram'.

Reviewer #2: Holt et al., described the epidemiology of animal and human brucellosis on dairy farms in rural Ludhiana, Punjab, India. The manuscript is well-written, and results are presented clearly and precisely. The findings from their study suggested a high seroprevalence of the disease in cattle, buffalo, and animal workers on farms in Punjab, as well as the significance of occupational exposure to the disease in this region. Hence, advocating for control and preventive strategies against production losses in animals and Brucella infection in animal workers.  

The authors made all the requested changes and corrections which have significantly improved the manuscript.

**Results**

-Does the analysis presented match the analysis plan?

-Are the results clearly and completely presented?

-Are the figures (Tables, Images) of sufficient quality for clarity?

Reviewer #1: Yes. The authors have revised the manuscript based on our suggestions.

Tables 6 and 7 can be presented as supplementary materials but I will let the editor make a final decision.

Reviewer #2: Yes

**Conclusions**

-Are the conclusions supported by the data presented?

-Are the limitations of analysis clearly described?

-Do the authors discuss how these data can be helpful to advance our understanding of the topic under study?

-Is public health relevance addressed?

Reviewer #1: Yes.

Reviewer #2: Yes

**Editorial and Data Presentation Modifications?**

Reviewer #1: Accept.

Reviewer #2: (No Response)

**Summary and General Comments**

Reviewer #1: The manuscript has substantially improved after revisions. Thank you to the authors for considering our suggestions and revising the manuscript.

Reviewer #2: The authors did a great job articulating the limitations of the study. They also provided pertinent recommendations for the effective control of brucellosis in rural Ludhiana, Punjab, which are potentially applicable to other endemic regions.

PLOS authors have the option to publish the peer review history of their article (what does this mean?). If published, this will include your full peer review and any attached files.

Reviewer #1: No

Reviewer #2: No

---

## [Editor Report · Acceptance letter]

15 Mar 2021

Dear Miss Holt,

We are delighted to inform you that your manuscript, "Epidemiology of brucellosis in cattle and dairy farmers of rural Ludhiana, Punjab," has been formally accepted for publication in PLOS Neglected Tropical Diseases.

Best regards,

Shaden Kamhawi

co-Editor-in-Chief

Paul Brindley

co-Editor-in-Chief
